# Can the Six-Minute Walk Test Be Used to Individualize Physical Activity Intensity in Patients with Breast Cancer?

**DOI:** 10.3390/cancers13225851

**Published:** 2021-11-22

**Authors:** Nicole Tubiana-Mathieu, Thibault Cornette, Stephane Mandigout, Sophie Leobon, François Vincent, Laurence Venat, Elise Deluche

**Affiliations:** 1Department of Medical Oncology, CHU de Limoges, 87000 Limoges, France; oncologie@chu-limoges.fr (N.T.-M.); thibault.cornette@unilim.fr (T.C.); sophie.leobon@chu-limoges.fr (S.L.); laurence.venat@chu-limoges.fr (L.V.); 2CAPTuR, EA3842, Faculty of Medicine, University Hospital, 87000 Limoges, France; 3Department of Physiology, University Hospital, 87000 Limoges, France; francois.vincent@unilim.fr; 4HAVAE, EA6310, Faculty of Science and Technology, University of Limoges, 87000 Limoges, France; stephane.mandigout@unilim.fr

**Keywords:** cardiopulmonary test, six-minute walk test, adapted physical activity, breast cancer, home training

## Abstract

**Simple Summary:**

Physical activity has proven to be effective in breast cancer patients. Appropriate exercise intensity for each patient is necessary to maintain this practice in patients with possible comorbidities and potential adverse events of specific treatments. These programs should be proposed to many patients so to prescribe the adapted program; this will necessitate easy and cost-effective tests. It is useful to use an adapted target heart rate (HR) to prescribe exercise intensity. In this work, we assessed the potential equivalence of the ventilatory threshold HR obtained during a cardiopulmonary exercise test and the HR measured over the last 3 min of the six-minute walk test (6MWT-HR). At baseline, the 6MWT-HR was in good agreement and showed moderate but statistical correlation with the VT-HR in breast cancer patients. The best correlation between these two tests was obtained after the APA program and chemotherapy. These results were independent of body mass index. The 6MWT is useful to prescribe APA programs before and also after chemotherapy concomitant with physical activity.

**Abstract:**

Background: Adapted physical activity (APA) aids breast cancer patients. It is necessary to use an adapted target heart rate (HR) when prescribing exercise intensity. Methods: In total, 138 patients previously included in two published randomized clinical trials underwent the CPET and 6MWT before and after adjuvant therapy. Of these patients, 85 had performed APA, and 53 had received only the usual therapy. HRs were recorded during the two tests. Results: Before starting chemotherapy, good agreement (intraclass correlation (ICC) 0.69; confidence interval at 95% IC_0.95_ (0.591–0.769); *p* < 0.001) and a moderate correlation were evident between the 6MWT-HR and ventilatory threshold HR of the CPET (r = 0.70; *p* < 0.001). Good agreement and a high positive correlation were noted only in the group who engaged in APA (ICC 0.77; IC_0.95_ (0.659–0.848); *p* < 0.001; r = 0.8; *p* < 0.01); moderate agreement and a moderate positive correlation were apparent in the control group (ICC 0.57; IC_0.95_ (0.329–0.74); *p* < 0.001; r = 0.6; *p* < 0.01). The correlations were independent of age and body mass index. Conclusions: The 6MWT-HR can be used to prescribe exercise intensity for breast cancer patients both before and after specific treatment with concomitant APA.

## 1. Introduction

Breast cancer is the predominant cancer in females worldwide. Various treatments improve survival, but the adverse events include a reduced quality of life, fatigue, and loss of muscular strength and aerobic fitness [1,2,3]. Physical activity (PA) reduces fatigue, depression, and anxiety and improves psychological and physiological functioning and thereby health-related quality-of-life, muscle strength, and cardiorespiratory fitness [4,5]. PA is recommended for all cancer patients immediately after diagnosis [6]. PA programs must be individualized to enhance adherence. The rate of adherence to adapted PA (APA) ranges from 60% to 70% for home activities [7,8] and from 59% to 98% for institutional activities [9,10]. In addition, we noticed in one of our previous studies (the SAPA trial) that exercise intensity and duration varied by the chemotherapy cycle [11].

Most recommended APA programs include endurance and resistance sessions [12]. Aerobic fitness, including heart rate (HR), must be evaluated when planning a program. Aerobic fitness can be assessed in various ways, with the gold standard (most accurate) being the cardiopulmonary exercise test (CPET) [13]. Training intensity is usually prescribed to the HR of the first ventilator threshold (VT) [14] of aerobic endurance when planning rehabilitation for cancer [11], diabetes [15], hemodialysis [16], and chronic obstructive pulmonary disease [17]. However, the CPET is expensive and requires a specific infrastructure and therapists. There is a need for a simple, reproducible, and inexpensive test yielding accurate physical fitness data that allow prescription of exercise intensity. The six-minute walk test (6MWT) seems to be the best candidate. This test is a simple field walking test used to give an objective indication of functional capacity and to enable respiratory rehabilitation [18]. The 6MWT is easy, fast, well tolerated by patients, and requires minimal human and financial resources. It is similar to daily activity. Some studies show good relationships between the HR or VO2 peak with 6MWT and CPET by comparing HR or VO2 values to target intensities [19,20]. Gayda et al. showed that the 6MWT cardiorespiratory requirement values did not differ from symptom-limited exercise test values at the ventilatory threshold except for ventilation [19]. However, in Kervio et al., HR and VO2 peak recorded during 6MWT were higher than that observed at the first VT in healthy elderly subjects (60–70 years) [21]. To the best of our knowledge, no study has really investigated whether there is a relationship between HR measured during a 6 min test and CPET in breast cancer patients.

Before treatment, breast cancer patients may not exercise because of advanced age, comorbidities, or a sedentary lifestyle. After treatment, exercise tolerance is reduced by adjuvant treatments and pathologies (anemia, cardiopulmonary pathologies, reduced cardiac output, and reduced muscle quality and quantity) [22]. As a measure of patient improvement, HR is simple to monitor via a pulse oximeter (for example) during APA. Here, we determined the extent of agreement between the ventilatory threshold HR (VT-HR) measured during the CPET and the HR measured during the last 3 min of the 6MWT (6WMT-HR) before and after adjuvant treatment. We used data on patients enrolled in our two published prospective studies that evaluated the impact of APA on peak oxygen uptake (VO_2peak_) values during breast cancer treatments [11,23].

## 2. Materials and Methods

### 2.1. Setting and Participants

Data were collected from the patients included in our two prospective trials, the SAPA (Clinicaltrials.gov—NCT01322412) [11] and APAC (Clinicaltrials.gov—NCT0179561) trials [23], both of which were conducted at Limoges University Hospital. Eligible patients were women aged 18–75 years with early-stage breast cancer treated with chemotherapy (neoadjuvant or adjuvant) followed by radiotherapy. All patients received the same chemotherapy. Six courses (three FEC 100, three docetaxel) were administered every 21 days, and trastuzumab was prescribed for 12 months if the tumor was positive for human epidermal growth factor receptor-2 (HER2). The exclusion criteria were symptomatic cardiac or pulmonary disease, metastatic disease, a left ventricular ejection fraction <50%, ongoing beta-blocker use, and/or a family history of sudden death in a first-degree relative. In both trials, half of all patients were randomized to very similar 6-month APA programs, and the other half received the usual care. The SAPA trial compared patients undergoing APA with others who were not; the APAC trial compared patients undergoing APA during, after, or both during and after, specific cancer treatment. In our present study we kept only the groups with APA and without APA during the first 6 months of treatment. This combined study was approved by the Ethics Committee of Limoges Hospital (approval no. 379-2020-35).

### 2.2. Study Design

We combined the relevant data for all patients from the SAPA and APAC trials. Both were open, interventional, single-center, prospective, randomized phase III trials.

The SAPA trial compared two groups:Group A performed 6-month home-based APA during adjuvant therapy.Group B received conventional management during adjuvant therapy.

The APAC trial compared three groups: Group A performed 6-month home-based APA during adjuvant or neoadjuvant therapy.Group B performed 6-month home-based APA after adjuvant or neoadjuvant therapy.Group C performed 12-month home-based APA during and after adjuvant or neoadjuvant therapy.

In the two trials, baseline was the start of chemotherapy whether it be neo or adjuvant therapy. The first follow up was the end of chemotherapy.

All patients received the same nutritional counselling. The APA targets were identical in the two studies. The CPET and 6MWT were performed at baseline (at the start of chemotherapy) and after 6 months of adjuvant treatment. Data were available for 138 patients. Data of the SAPA trial were obtained in June 2013 and those from the APAC trial in November 2019. Of the 138 patients, 53 did not perform APA during adjuvant treatment (because they were randomized out), and 85 performed APA during adjuvant treatment.

### 2.3. Study Outcomes

The primary objective was the extent of agreement between the VT-HR of the CPET and the 6WMT-HR. The secondary objectives included the correlation between these two values, their correlations with age-based predicted maximal HR (HR_max_), and the effects of age and body mass index (BMI) on the values.

HR_max_ was calculated using the Gellish formula [24].

### 2.4. The Tests Used in the SAPA and APAC Trials

#### 2.4.1. Cardiopulmonary Exercise Test

All patients underwent the CPET to evaluate VO_2peak_, HR_peak_, and VT-HR. The CPET followed the clinical guidelines for cancer patients [25]. A 12-lead electrocardiographic monitoring system (Corina, GE Medical Systems IT Inc., Milwaukee, WI, USA) was used during evaluation. CPETs were performed on an electronically braked cycle ergometer. Peak oxygen uptake was highest during exercise. The VT-HR and the HR_peak_ (maximum HR during the CPET) were recorded. The first VT was defined as being when CO_2_ production began to increase disproportionately to O_2_ consumption, as measured using the Wasserman method [26]. This test was supervised by a respiratory physiologist (more details in previous work) [23].

#### 2.4.2. Six-Minute Walk Test

The 6MWT was performed along a 25 m segment of a silent corridor [11,27]. Patients were briefed to walk as quickly as possible for up to 6 min. Patients were orally notified after each minute passed (e.g., “Four minutes left”). The HR during the last 3 min was the 6MWT-HR and the total distance walked were recorded.

#### 2.4.3. Body Composition

The BMI was calculated as mass (kg)/height (m^2^).

### 2.5. Exercise Training

Both the SAPA and APAC trials featured the same APA program. An exercise specialist provided detailed patient-specific information and evaluated the home activity and patient-fitness levels during each chemotherapy course. The home-based program combined aerobic and resistance sessions. Aerobic exercises were performed at least twice weekly on a bicycle ergometer. For exercise intensity, the HR-VT was used [28]. Exercise duration increased from 20 to 40 min over the program. Patients were encouraged to go for brisk walks in addition to bicycling. Resistance sessions were performed weekly using elastic bands, targeting the abdominal, hamstring, quadriceps, triceps, and surae/gluteus maximus muscles. Each resistance training session featured two sets of 8–12 repetitions.

### 2.6. Statistical Analysis

All data were collected and analyzed using STATVIEW software ver. 5.0 (SAS Institute, Cary, NC, USA) and R software ver. 3.5.3 (R Foundation for Statistical Computing, Vienna, Austria). Quantitative results were expressed as means ± standard deviation or as medians (with ranges) and qualitative results were expressed as numbers with percentages. Continuous outcomes were compared using the Student *t*-test or the non-parametric Mann–Whitney U-test, as appropriate. We determined the Pearson correlation coefficients. Intraclass correlations (ICCs) were used to measure the extent of agreement between continuous outcomes. Correlation refers to the presence of a relationship between two different variables, whereas agreement looks at the concordance between two measurements of one variable. In our study we wanted to evaluate the correlation between VT-HR during CPET and 6WMT-HR. However, a good correlation does not imply a good concordance of measurement. It was therefore important to perform a Bland and Altman test and to determine the ICC to evaluate this agreement. Based on the 95% confidence interval, values of <0.5, 0.5–0.75, 0.75–0.9, and >0.90 were indicative of poor, moderate, good, and excellent reliability, respectively [29]. Statistical significance was set at *p* < 0.05.

## 3. Results

### 3.1. Patient Characteristics at Baseline

In total, 138 patients were enrolled (44 from the SAPA study and 94 from the APAC study) (Figure 1). The baseline patient characteristics are listed in Table 1. The average age was 50 years (range 29–74 years). Half of all patients exhibited a normal BMI, but 46 were overweight. Stage II breast cancer was the most frequent cancer. The frequency of self-reported cardiovascular risk factors was low. All patients received chemotherapy, and 93% received adjuvant radiotherapy. It was noted that patients’ characteristics were not different between the populations of the two trials

### 3.2. HR at Baseline

The results are shown in Table 2. The mean VT-HR was 127.8 ± 14.0 bpm, thus being 83.2% of the HR_peak_ obtained during the same test and 74.3% of the HR_max_. The mean 6MWT-HR was 129.3 ± 15.5 bpm (75.2% of the HR_max_). The effects of age and BMI on the VT-HR and 6MWT-HR are shown in Table 3. 6MWT-HR and VT-HR were weakly but significantly (inverse) associated with age but were not associated with BMI.

### 3.3. Relationship between the 6MWT-HR and VT-HR at Baseline

The VT-HR and 6MWT-HR were highly correlated (r = 0.7; *p* < 0.0001) (Table 3). The Bland–Altman plot revealed moderate agreement between the VT-HR and 6MWT-HR (ICC 0.69; IC_0.95_ (0.591–0.769); *p* < 0.001) (Figure 2). Comparison using this plot analysis suggests the 6MWT compared to its gold standard could be used interchangeably, with a few precautions though. No impact of age or BMI on this relationship was found.

### 3.4. Impact of APA during Adjuvant Treatment on the 6MWT-HR and VT-HR

Eighty-three patients with available baseline data were enrolled in the APA program. Of these patients, only 75 underwent repeats of all tests after specific adjuvant treatment; 43 control group patients were evaluated at the same time. The 6MWT-HR and VT-HR after APA (131.2 ± 17.0 and 129.4 ± 13.2 bpm, respectively) were similar to those at baseline (129.3 ± 16.1 and 127.5 ± 14.2 bpm, respectively), despite chemotherapy. After treatment, the HR_peak_ in the APA group increased significantly (baseline vs. post-treatment: 153.5 ±15.7 vs. 156.6 ± 17.1 bpm, *p* = 0.0439), whereas that in the control group decreased significantly (155.1 ± 15.7 vs. 152.3 ± 13.6 bpm, *p* = 0.008).

After treatment, in the APA group, a significant positive correlation and good concordance were evident between the 6MWT-HR and VT-HR (r = 0.8; *p* < 0.001 and ICC 0.77; IC_0.95_ (0.659–0.848); *p* < 0.001). In the control group, a significant moderate positive correlation and moderate concordance were evident between the 6MWT-HR and VT-HR (r = 0.6; *p* > 0.001 and ICC 0.57; IC_0.95_ (0.329–0.74); *p* < 0.001).

## 4. Discussion

We sought a simple reproducible test that could be used to prescribe the exercise intensity of an APA program in women with breast cancer. One major result of our study was the good agreement (ICC = 0.69) and moderate correlation (r = 0.70) between the 6MWT-HR and VT-HR prior to chemotherapy. Thus, the HR during the last 3 min of the 6MWT allows calculation of the required exercise intensity. To the best of our knowledge, these results are novel. In published trials, exercise intensity and the methods used to monitor exercise parameters vary substantially. A submaximal 6MWT result is regularly used to diagnose chronic heart failure and chronic obstructive pulmonary disease, but the test has rarely been used to evaluate breast cancer patients after APA and adjuvant chemotherapy [30]. Exercise intensities are set using different methods, including percentage of the HR_max_ (predicted range 40–80%) [31,32,33], the Karvonen method, the HR eliciting a blood lactate concentration of 2–3 mmol, the HR associated with a set percentage of the directly measured maximal oxygen uptake (range 45–75%) [9,34], and subjective measures (low-to-moderate, moderate, or moderate-to-vigorous intensity) [35]. Thus, no consensus has emerged on how to determine optimal exercise intensity.

When planning APA, the aim is to define a target intensity appropriate for individual physical capacity. However, our study showed that the 6-month CPET HR_peak_ can change significantly (with or without an exercise program). It is essential to carefully (and regularly) adjust the exercise intensity to reduce fatigue and to enhance APA program effectiveness. We showed that the 6MWT identifies an appropriate HR target. The 6MWT is very simple, and it is possible to individualize exercise intensity at all times.

Our second major finding was the good agreement between the 6MWT-HR and VT-HR in patients on the APA program, but not in the controls. After treatment, the 6MWT-HR and VT-HR were stable in both groups. The CPET HR_peak_ (only) increased significantly in the APA group and decreased in the control group after exercise, highlighting the positive effects of exercise during chemotherapy. The agreement between 6MWT-HR and VT-HR was high in the APA group, but only moderate in the control group, as reported previously [36]. Control group deconditioning was in play. After rehabilitation, Morard et al. [37] reported similar results in patients with coronary disease but not in a healthy elderly population or in chronic heart-failure patients, as also shown by Kervio et al. [38]. Any useful role played by APA intensity in breast cancer patients remains debatable. Higher-activity intensity improved survival in the Women’s Health Initiative (WHI) study [39], but not in another study [40].

In many studies, the target training intensity was 60–80% of the HR_max_. In the present study, the 6MWT-HR and VT-HR were approximately 75% of the HR_max_. This target was probably adapted to the exercise programming because our results were good. In both previous studies (SAPA and APAC), the 6MWT was not used to plan APA intensity, but rather to compare walking capacity before and after chemotherapy in both the APA and control groups. The walking distance in the APA group increased significantly from 521.6 ± 61.2 to 539.1 ± 59.9 m (*p* = 0.0011), whereas that of the control group decreased significantly from 524.6 ± 8.4 to 514.9 ± 53.1 m (*p* = 0.03). However, both groups exhibited an increase in the VO_2peak_. The CPET showed that the VO_2peak_ in the APA group after APA and chemotherapy increased from 16.2 ± 3.4 to 22.6 ± 5.2 mL·kg^−1^·min^−1^ (*p* = 0.009), whereas that of the control group decreased from 21.8 ± 4.8 to 20.4 ± 4 mL·kg^−1^·min^−1^ (*p* = 0.0003) [11,23]. These data were our trial objectives.

Regarding the limitations of the two trials, our patient recruitment was biased since only volunteers were recruited, and approximately 20% of patients refused the APA program. Additional limitations include secondary data analyses with retrospective data. Only patients who could tolerate chemotherapy and were not on beta-blockers were included. Moreover, the patient profile was heterogeneous, such as the wide-age range of 29–74 years, and the presence of comorbidities such as obesity might have affected the results. However, we found that age was inversely correlated with the 6MWT-HR and VT-HR but did not affect the relationship between the two values. The age-dependence is commonly known but has been proven only in patients aged 40–65 years. We obtained the same results in patients aged 29–74 years. Moreover, BMI did not affect either value or the relationship between the 6MWT-HR and VT-HR.

## 5. Conclusions

PA clearly benefits breast cancer patients. However, maintenance of appropriate exercise intensity is a real challenge for patients with comorbidities or those who experience adverse treatment effects. The 6MWT-HR may aid prescription of an appropriate training intensity.

## Figures and Tables

**Figure 1 cancers-13-05851-f001:**
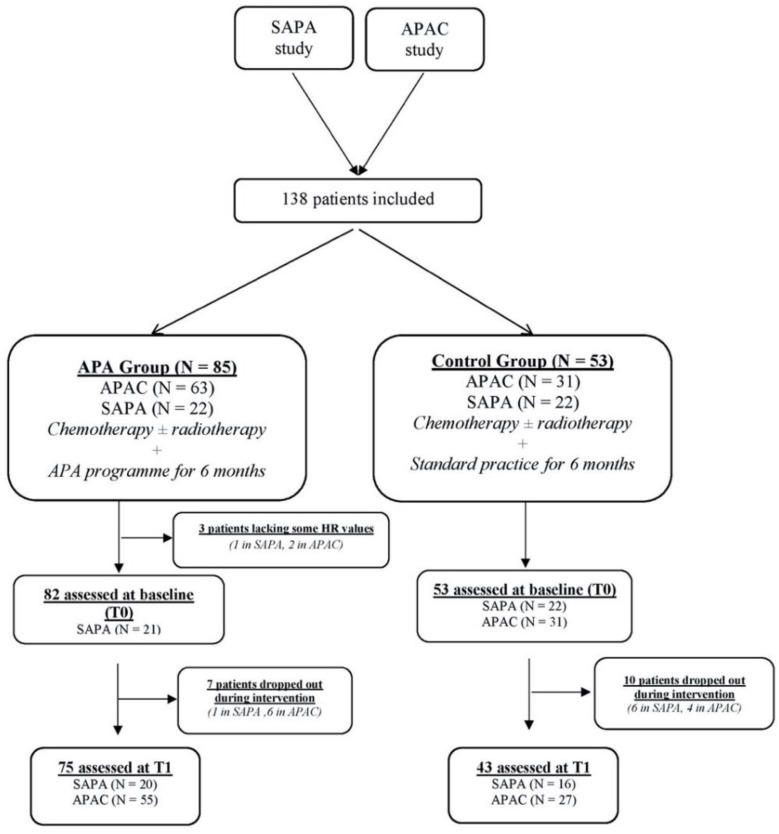
Study flowchart.

**Figure 2 cancers-13-05851-f002:**
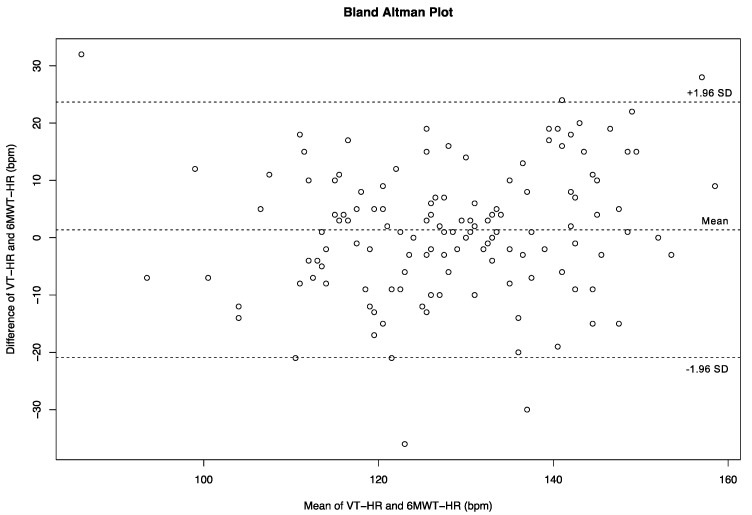
The extent of agreement between the 6MWT-HR and VT-HR at baseline (Bland–Altman plot).

**Table 1 cancers-13-05851-t001:** Patient characteristics at baseline (*N* = 138).

Patient Characteristics	All Patients (*N* = 138)	APA Group (*N* = 85)	Control Group (*N* = 53)
Age (years), median (min–max)	52 (29–74)	51 (29–74)	50 (37–72)
BMI (kg/m²), mean ± SD	25.6 ± 4.8	25.4 ± 5.2	25.8 ± 4.2
BMI (kg/m²), *n* (%)			
Thin (<18.5)	3 (2.2)	3 (3.5)	3 (2.2)
Normal (≥18.5, <25)	70 (50.7)	44 (51.8)	70 (50.7)
Overweight (≥25, <30)	46 (33.3)	27 (31.8)	46 (33.3)
Obese (≥30)	19 (13.8)	11 (12.9)	19 (13.8)
Cancer stage, *n* (%)			
I	23 (16.8)	19 (22.6)	23 (16.8)
II	97 (70.8)	54 (64.3)	97 (70.8)
III	17 (12.4)	11 (13.1)	17 (12.4)
Bilateral, *n* (%)	6 (4.5)	4 (4.8)	2 (4.0)
Mastectomy, *n* (%)	40 (29.9)	25 (29.8)	15 (30.0)
Lumpectomy, *n* (%)	95 (70.9)	60 (71.4)	35 (70.0)
Axillary dissection, *n* (%)	70 (51.1)	42 (49.4)	28 (53.8)
Comorbidities, *n* (%)	22 (15.9)	13 (15.3)	9 (17.0)
Hypertension	19 (13.8)	12 (14.1)	7 (13.2)
Stroke	2 (1.5)	0 (0)	2 (3.7)
Phlebitis	2 (1.5)	1 (1.2)	1 (1.9)
Hemoglobin (g/L), mean ± SD	12.8 ± 0.8	12.7 ± 0.9	13.3 ± 0.4

**Table 2 cancers-13-05851-t002:** The HRs of the CPET and 6MWT at baseline (*N* = 138).

Tests Characteristics	No. of Patients	Mean	SD	Range
6MWT				
6MWT-HR (bpm)	137	129.3	15.5	90.0–171.0
CPET				
VT-HR (bpm)	136	127.8	14.0	70.0–162.0
HR_peak_ (bpm)	137	154.1	15.7	107.0–187.0
Calculated variables				
Predicted HR_max_ (bpm)	138	172.0	8.0	153.2–185.6
% VT-HR/predicted HR_max_	136	74.3	7.6	40.7–95.5
% 6MWT-HR/predicted HR_max_	137	75.2	8.5	57.2–97.3
% VT-HR/HR_peak_	136	83.2	6.4	47.0–107.3
% 6MWT-HR/HR_peak_	136	84.3	8.6	64.4–111.9

**Table 3 cancers-13-05851-t003:** Correlations between physiological parameters and HRs at baseline (*N* = 138) and between the HRs of the CPET and 6MWT.

Tests Characteristics	r	IC_0.95_	*p*-Value
6MWT-HR vs. age	−0.37	(−0.50, −0.21)	<0.001
6MWT-HR vs. BMI	−0.12	(−0.28, −0.05)	0.170
VT-HR vs. age	−0.38	(−0.52, −0.23)	<0.001
VT-HR vs. BMI	−0.12	(−0.29, 0.04)	0.149
6MWT-HR vs. VT-HR	0.70	(0.60, 0.77)	<0.001

## Data Availability

Data are available from the corresponding author upon reasonable request.

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
