# Peer review of "Can the Six-Minute Walk Test Be Used to Individualize Physical Activity Intensity in Patients with Breast Cancer?"

_cancers, 2021, doi:10.3390/cancers13225851_

Round 1

Reviewer 1 Report

The study combines data from two previous RCTs that examined the impact of adapted physical activity on patients with breast cancer. The objective of this secondary analysis was to examine if prescribing intensity of adapted physical activity programs can be measured using a 6minute walk test (6MWT) as opposed to the most cumbersome gold standard (CPET).

Specific comments:

Introduction (lines 59-63). The authors should expand on why the 6MWT is the best candidate for assessing physical fitness and allow prescription exercise? To what kind of clinical populations the test has been used? What strenght of correlations were achieved between the 6MWT and the gold standard of CPET? What were the differences among studies that showed a good correlation and those that did not (refs 19, 20, 21). If there is ambiguity in the prescriptive value of the 6MWT, how using it with patients with breast cancer help resolve these ambiguities?

Methods

Line 95: when was data collected for each trial? this information helps the reader to understand the greater context

Lines 106 - 110: Can you please clarify, what was the baseline for the APAC trial that included patients during adjuvant and neoadjuvant treatment? In other words, how far apart were the baseline and follow-up measures in the two trials?

Line 170. The authors mention that "both trial populations were homogeneous." This statement should probably be edited to clarify that populations were heterogeneous within each trial (as they also acknowledge in their limitations) but similar between the two trials.

Discussion:

Lines 248-254. It is not clear the purpose of present this data. This information appears to be relevant to original trials, but does not seem to be relevant to the question of this article: can we substitute CPET with 6MWT to prescribe exercise intensity to patients with breast cancer?

Lines 255-264: the limitations presented here apply to the original studies but not necessarily to the current analyses. Additional limitations include secondary data analyses with retrospective data and potentially a different time frame of follow-up between the two studies. 

Minor edits:

Simple Abstract: the language needs to be simplified

Scientific Abstract: the authors need to clarify that the two groups were based on randomization

Author Response

Reviewer 1:

According to your letter of and your advice, we have revised the paper and implemented the remarks. Please find our point-by-point responses in this accompanying rebuttal letter and appropriate changes have been implemented in our revised text. We are really grateful to the reviewer 1 for their excellent suggestions and critical reading of the manuscript, which helped us to improve the original study.

Regarding our responses to the specific points raised by the reviewer 1, the reviewer’s comments appear in black font, our responses appear in blue font. In the paper, modifications have been highlighted in red.

The authors should expand on why the 6MWT is the best candidate for assessing physical fitness and allow prescription exercise? To what kind of clinical populations the test has been used? What strenght of correlations were achieved between the 6MWT and the gold standard of CPET? What were the differences among studies that showed a good correlation and those that did not (refs 19, 20, 21). If there is ambiguity in the prescriptive value of the 6MWT, how using it with patients with breast cancer help resolve these ambiguities?

Response : To follow reviewer’s comments, we modified paragraph and the 6MWT interest has been developed in the text (L59-70)

This test has been validated in patients with chronic heart failure, chronic obstructive pulmonary disease, coronary artery disease but also in elderly healthy patients. We have described in the text.

The relationship between the two tests has been corrected in the text L63-69

The ambiguity possible is based on different populations studied in literature. The aim of our study is the proposition to perform this test in a large population with breast cancers in which APA is proved efficient.

 Methods

Line 95: when was data collected for each trial? this information helps the reader to understand the greater context

Line 117-118 : To follow reviewer’s comments, we added : “Data of SAPA trial were obtained in June 2013 and those from APAC trial in November 2019.”

Lines 106 - 110: Can you please clarify, what was the baseline for the APAC trial that included patients during adjuvant and neoadjuvant treatment? In other words, how far apart were the baseline and follow-up measures in the two trials?

Line 113 We appreciate the reviewer’s suggestion, and we added this sentence : “In the two trials baseline was the start of chemotherapy whether it be neo or adjuvant therapy. First follow up was the end of chemotherapy.

Line 170. The authors mention that "both trial populations were homogeneous." This statement should probably be edited to clarify that populations were heterogeneous within each trial (as they also acknowledge in their limitations) but similar between the two trials

Line 181: We wanted to show that the populations included in the trials were not different with similar, median age BMI, antecedent, treatment and APA program. To follow reviewer’s comments, we have corrected in the text : “It is noted that patients ‘characteristics were not different between the population of the two trials”

Discussion:

Lines 248-254. It is not clear the purpose of present this data. This information appears to be relevant to original trials, but does not seem to be relevant to the question of this article: can we substitute CPET with 6MWT to prescribe exercise intensity to patients with breast cancer?

We appreciate the reviewer’s comments. We agree that these data were not relevant with the purpose of article but seemed necessary to prove efficacy of exercise program proposed to patients and may explain the data found at the end of APA program.

Lines 255-264: the limitations presented here apply to the original studies but not necessarily to the current analyses. Additional limitations include secondary data analyses with retrospective data and potentially a different time frame of follow-up between the two studies. 

We agree. The cited limitations were not attributed to the current analyses but to the two first clinical studies.  We modified this sentence Line 265 and added “Additional limitations include secondary data analyses with retrospective data”

We disagree because the frame of follow up was not different on the two clinical studies.

Simple Abstract: the language needs to be simplified

To follow reviewer’s comments, Corrections have been performed

Scientific Abstract: the authors need to clarify that the two groups were based on randomization

To follow reviewer’s comments, Corrections have been performed

Reviewer 2 Report

The authors compare the heart rate in the last three minutes of the six-minute walk test (6MWT) with the HR of the first ventilator threshold from the gold standard cardiopulmonary exercise test to determine their relationship in order to tailor exercise programs for patients with breast cancer. The conclusions are appropriately tempered to recognize that the tests are correlated enough to suggest their relationship may allow substitution.  I do think the concept of offering a more affordable HR assessment for cancer patients is an important contribution. I suggest some changes below that may strengthen the manuscript and its conclusions.   

Introduction

There is a detailed background provided on the importance of exercise programs for patients with breast cancer and the use of the CPET as the gold standard. However, I found the justification for the 6MWT lacking. I noticed lines 216-217 of the discussion stating “A submaximal 6MWT result is regularly used to diagnose chronic heart failure and chronic obstructive pulmonary disease, but the test has rarely been used to evaluate breast cancer patients after APA and adjuvant chemotherapy [30] “ provide such needed additional justification for the reader to appreciate the choice of this particular test and would recommend moving that sentence into the sentence of line 60 in the introduction.

Methods

2.1 Setting/Participants

Line 88-90 was slightly difficult to follow without re-reading. As I understood, both trials have patients randomized to a physical activity program or usual care. When stating the SAPA trial compared patients undergoing the program to “others who were not”, I believe this should be consistent with the prior sentence and state with the group receiving usual care. When moving to the following section on study design, however, it appears that the APAC trial was entirely a comparison of physical activity, but at different points in the receipt of adjuvant or neoadjuvant therapy, so usual care is not a comparison group, which needs to be clarified. Also, if both trials are prospective, when is this comparison being done for the SAPA trial? The APAC trial states during, after, and both during and after a specific cancer treatment. But it is previously stated all received the same chemotherapy, what specific cancer treatment are you referring to?

2.3 Study outcomes

It would be useful for any reader to understand the value of agreement vs correlation and why one is primary vs secondary objective. A short statement explaining that strong correlation does not always represent agreement and therefore the importance of the difference plot to convey the correspondence/discrepancy between the two tests is needed.

Results

Study Flowchart

For some reason the flowchart only has only text and no lines/arrows indicating separations. Perhaps there was an error in uploading or file format.

Section 3.3. and Figure 2.

This section is the primary objective for the study. This would be better served if the results added a statement that reflected on the moderate agreement, such as by stating that comparison using this plot analysis suggests the 6MWT compared to its gold standard can be used interchangeably. Also, I feel the lines representing the mean and limits of agreement on the Bland-Altman plot should be labeled.

Author Response

Reviewer 2:

According to your letter of and your advice, we have revised the paper and implemented the remarks. Please find our point-by-point responses in this accompanying rebuttal letter and appropriate changes have been implemented in our revised text. We are really grateful to the reviewer 1 for their excellent suggestions and critical reading of the manuscript, which helped us to improve the original study.

Regarding our responses to the specific points raised by the reviewer 1, the reviewer’s comments appear in black font, our responses appear in blue font. In the paper, modifications have been highlighted in red.

Introduction

There is a detailed background provided on the importance of exercise programs for patients with breast cancer and the use of the CPET as the gold standard. However, I found the justification for the 6MWT lacking. I noticed lines 216-217 of the discussion stating “A submaximal 6MWT result is regularly used to diagnose chronic heart failure and chronic obstructive pulmonary disease, but the test has rarely been used to evaluate breast cancer patients after APA and adjuvant chemotherapy [30] “ provide such needed additional justification for the reader to appreciate the choice of this particular test and would recommend moving that sentence into the sentence of line 60 in the introduction.

Thank you for this comment. The introduction has been corrected to justify using of 6MWT (Line 62-70) : “ Some studies show good relationships between the HR or VO2 peak  with 6MWT and CPET by comparing HR or VO2 values to target intensities [19-20]. Gayda et al. show that the 6MWT cardiorespiratory requirement values did not differ from symptom-limited exercise test values at ventilatory threshold except for ventilation [19]. However in Kervio et al HR and VO2 peak recorded during 6MWT were higher than that observed at the first VT in healthy elderly subjects( 60-70y)[21]. To the best of our knowledge, no study has really investigated whether there is a relationship between HR measured during a 6-minute test and CPET in breast cancer patients.”

Methods

2.1 Setting/Participants

Line 88-90 was slightly difficult to follow without re-reading. As I understood, both trials have patients randomized to a physical activity program or usual care. When stating the SAPA trial compared patients undergoing the program to “others who were not”, I believe this should be consistent with the prior sentence and state with the group receiving usual care. When moving to the following section on study design, however, it appears that the APAC trial was entirely a comparison of physical activity, but at different points in the receipt of adjuvant or neoadjuvant therapy, so usual care is not a comparison group, which needs to be clarified. Also, if both trials are prospective, when is this comparison being done for the SAPA trial? The APAC trial states during, after, and both during and after a specific cancer treatment. But it is previously stated all received the same chemotherapy, what specific cancer treatment are you referring to?

Both trials were prospective, but APAC was performed after SAPA (dates of data was specified in text). All patients received the same chemotherapy 3 courses of FEC (5FU, EPIRUBICINE CYCLOPHOSPHAMIDE) followed by 3 courses of TAXOTERE.

In the 2 trials we have a control group; in SAPA trial patients were randomized APA program versus not and in APAC trial, patients randomized in arm B do not perform APA during chemotherapy but only after chemotherapy, so after first follow up.

Line 97: We appreciate the reviewer’s suggestion and we added on the text: “In our present study we kept only the group with APA and without APA during the first 6 months of the APAC Trial”

2.3 Study outcomes

It would be useful for any reader to understand the value of agreement vs correlation and why one is primary vs secondary objective. A short statement explaining that strong correlation does not always represent agreement and therefore the importance of the difference plot to convey the correspondence/discrepancy between the two tests is needed.

To follow reviewer’s comments, we proposed in the text (Line 171-176): “Correlation refers to the presence of a relationship between two different variables, whereas agreement looks at the concordance between two measurements of one variable. In our study we want to evaluate the correlation between VT-HR during CPET and 6WMT-HR. However, a good correlation does not imply a good concordance of measurement. It is therefore important to perform a Bland and Altman test and to determine the ICC to evaluate this agreement.”

Results

Study Flowchart

For some reason the flowchart only has only text and no lines/arrows indicating separations. Perhaps there was an error in uploading or file format.

We apologise for the inconvenience and have corrected the problem which must be due to the change of software between the computers.

Section 3.3. and Figure 2.

This section is the primary objective for the study. This would be better served if the results added a statement that reflected on the moderate agreement, such as by stating that comparison using this plot analysis suggests the 6MWT compared to its gold standard can be used interchangeably. Also, I feel the lines representing the mean and limits of agreement on the Bland-Altman plot should be labeled.

Thank you for these comments. As your comment for the figure, we add the mean and limit. Moreover, we add on this section 3.3 a sentence to emphasize with the primary outcome of the study.

Line 201-203: “Comparison using this plot analysis suggests the 6MWT compared to its gold standard could be used interchangeably, with a few precautions though.”
